# Active Learning Helps Pretrained Models Learn the Intended Task

**Alex Tamkin**[*] **Dat Nguyen**[*] **Salil Deshpande**[*] **Jesse Mu** **Noah Goodman**

Stanford University

## Abstract

Models can fail in unpredictable ways during deployment due to *task ambiguity,* when multiple behaviors are consistent with the provided training data. An example is an object classifier trained on red squares and blue circles: when encountering blue squares, the intended behavior (classifying shape vs. color) is ambiguous. We investigate whether pretrained models are better active learners, capable of **choosing examples that improve robustness** to such spurious correlations and domain shifts. Intriguingly, we find that better active learning is an emergent property of the pretraining process: pretrained models require **up to 5× fewer labels** when using uncertainty-based active learning, while non-pretrained models see no or even negative benefit. We find these gains come from an ability to select examples with attributes that disambiguate the intended behavior, such as rare product categories or atypical backgrounds. These attributes are far more linearly separable in pretrained model's representation spaces vs non-pretrained models, suggesting a possible mechanism for this behavior. Code and training scripts are available at: https://github.com/alextamkin/active-learning-pretrained-models.

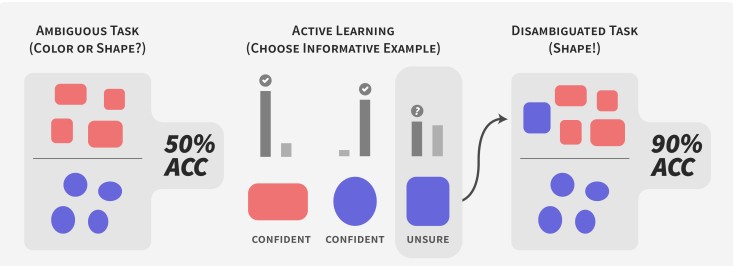

Figure 1: **Active learning can resolve task ambiguity in datasets.** Here, the provided training data leaves the model unsure of the intended task: is it to predict the shape or the color of the object? Pretraining enables models to identify and weigh various rich features, eliciting labels from informative examples (e.g. blue squares) that clarify the user's intention.

## 1   Introduction

Modern pretrained models can be adapted to new tasks with remarkably little data, enabling downstream applications for tasks with only tens or hundreds of examples [8, 46]. However these low-data applications magnify a fundamental problem in machine learning—namely, that datasets are often incomplete proxies for the desired behavior. For example, a sentiment classifier may behave

---

[*]Equal contribution. Correspondence: atamkin@stanford.edu

36th Conference on Neural Information Processing Systems (NeurIPS 2022).

unpredictably during the holidays if its training data lacks examples of toy reviews. Likewise, an object classifier may struggle to classify objects in atypical environments (e.g. camels in the Arctic) if the training data does not make it adequately clear which features are salient for the task. While these dataset flaws can be solved by labeling better examples, identifying such examples can be challenging, especially as the nature of these flaws may not be known in advance.

One way to describe these kinds of challenges is through the lens of *task ambiguity*: the failure of the training data to fully specify the user's intended behavior for all possible inputs. We consider whether pretrained models can automatically resolve their own task ambiguity through *active learning* (AL). In principle, AL allows models to resolve task ambiguity by identifying examples whose labels would be informative; for example, in Figure 1, the provided training data only contains red squares and blue circles, leaving the intended behavior for blue squares ambiguous. Asking for labels of blue squares resolves this ambiguity. This kind of interaction could clarify the intended behavior for different kinds of examples without the expectation that model developers must anticipate all potential gaps in the model's abilities.

However, AL has often seen limited success in practice. In traditional settings with smaller, non-pretrained models, several problems limit the effectiveness of AL, including label noise, examples that are challenging for models to learn, and a lack of generalizability of AL heuristics [38, 30]. But pretrained models possess several strengths which may enable them to overcome these challenges: they extract high-level semantic features, such as shape and color [44, 22], which can be used to identify informative examples, and they produce more calibrated uncertainty estimates which can be used for selecting ambiguous inputs [24, 14]. Moreover, in many real-world problems some data points are much more informative than others [21], suggesting the potential utility of AL in practical few-shot learning applications.

We consider the use of active learning (AL) on a range of real-world image and text datasets where task ambiguity arises. We compare several AL acquisition functions against a random-sampling baseline, and compare the difference in performance with and without the use of pretrained models. Our **contributions** demonstrate that:

1. Pretrained models trained with AL can select examples that resolve task ambiguity in the finetuning data
2. The resulting accuracy gains can be quite large in practice: up to $5\times$ reduction in labeled data points for the same performance, or +11% absolute gain for the same labeling budget
3. This ability to actively learn is an emergent property of pretraining: AL has a neutral or even harmful effect on non-pretrained models

## 2  Method

We study the pool-based active learning (AL) setup common in the literature [52], where we have a (possibly pretrained) model $\mathcal{M}$, a small *seed set* of training data $\mathcal{S} = \{(x_i, y_i)\}$, and a larger *pool* of unlabeled data $\mathcal{P} = \{x_i\}$. The AL procedure proceeds as follows: first, finetune $\mathcal{M}$ on $\mathcal{S}$ until convergence; then, select points $x_i$ from $\mathcal{P}$ that are deemed *most informative* according to an acquisition function $a(x; \mathcal{M})$, obtaining the corresponding labels $y_i$, until some budget $k$ of data points is exhausted. This newly labeled batch $\mathcal{B} = \{(x_i, y_i)\}$ is then added to the existing data $\mathcal{S}$, and the original model is retrained on $\mathcal{S}$ for the next acquisition step. This process is repeated for a fixed number of acquisition steps.

For our acquisition function, we adopt the classic *uncertainty sampling* approach to AL, in particular the *least confidence* heuristic, where we acquire points for which our model is *least confident* in its predicted label [52]. Specifically, treating the outputs of the model $\mathcal{M}$ as a probability distribution[1] over possible labels $p(y \mid x; \mathcal{M})$, we define the acquisition function to be

$$a(x; \mathcal{M}) = -\max_i p(y_i \mid x; \mathcal{M}) \tag{1}$$

This *least confidence* heuristic has shown to be simple and effective in a variety of settings [52, 23, 40], and we similarly find good results here (we refer to least confidence sampling as "uncertainty

---

[1]While in general there is no guarantee that this probability distribution will be well-calibrated, recent work has found that pretraining improves model calibration across a variety of settings, including on out-of-domain data [14, 24].

sampling" except in Section 5.2 where we explore other uncertainty-based acquisition functions including *entropy* and *margin* sampling).

On top of this standard AL pipeline, we propose the following change to improve the practical applicability of pretrained models in data-scarce settings:

**Removing the need for a separate validation set**  The AL cycle begins by finetuning the pretrained model on the seed set. If the size of the seed set is large enough, the seed set may be partitioned into a training set and a validation set, and early stopping may be performed on the validation set. However, in few-shot settings, labeling costs may be high, and the seed set may be too small to meaningfully partition.

Instead of an arbitrary fixed number of finetuning steps, we propose an alternative method to terminate finetuning in the absence of a validation set. Specifically, we find that a simple but effective heuristic is to stop finetuning when the training loss decreases to 0.1% of the original training loss at the start of finetuning. In our experiments, this heuristic performs as well as early stopping on an actual validation set (see Appendix D for more details). We also leverage standard learning rates and other hyperparameters recommended by model developers (see Appendix B).

By using a standardized recipe across tasks and removing the need for a separate validation set, our AL pipeline is more robust to the real-world difficulties of deploying AL where use of a validation set is impractical [38, 45], although further work is needed to capture the full extent of this recipe's generalizability.

## 3  Datasets

We consider a variety of datasets where task ambiguity manifests through a scarcity of particular kinds of examples. We consider two such kinds of examples: those defined by combinations of causal and spurious features (typical vs atypical backgrounds) as well as those defined by unseen attributes that shift during deployment (product categories and camera trap locations). These datasets provide an empirical testbed for the ability of pretrained models to choose disambiguating examples using active learning (AL).

### 3.1  Distinguishing causal from spurious features

Spurious correlations arise when multiple features are predictive of the label in a training dataset, yet it is ambiguous which ones are causally linked to the task label [21]. We consider two such datasets, and see whether AL can choose the *disambiguating examples* where the spurious features are not copresent with the causal features:

**Waterbirds**  The Waterbirds dataset [49] consists of photographs of landbirds or waterbirds digitally edited onto land or water backgrounds. The task is to classify whether the bird is a landbird or a waterbird. In the train set, 77% of the pictures feature landbirds and 23% waterbirds. 95% of both landbirds and waterbirds appear on land and water backgrounds, respectively. In the validation and test sets, this percentage is decreased to 50%, instead. Thus, the image background is a spurious feature the model may come to rely on when making the prediction.

**Treeperson**  As the Waterbirds dataset was synthetically generated, we also consider a dataset where we perform classification over real, unedited images with spuriously correlated objects. We use the object annotations in Visual Genome [33] to create a new dataset of 8,638 images called Treeperson, for which the task is to predict whether a person is in a given image. While 50% of the images contain a person in this dataset, each image also contains either a tree or a building, and the presence of these objects is spuriously correlated with the presence of people. At train time, 90% of training images with people contain a building, while 90% of training images without people contain a tree. Thus, a model may be incentivized to form representations that classify according to the presence of trees and buildings, rather than the presence of the actual causal variable of interest (people). These values are changed to 50% at test time, removing this correlation to evaluate how well the model learned the actual task of interest. For more details on this dataset, see Appendix C.

## 3.2 Measuring robustness to distribution shift

Distribution shifts occur when algorithms are evaluated on different data distributions than the ones they were trained on. Examples include changing the location or time of day that photos were taken, or changing the topic or author of a particular textual source. These shifts can reduce performance, and we consider whether AL can help choose diverse, informative examples that clarify how the model should behave over a range of natural distribution shifts.

**iWildCam2020-WILDS** This dataset considers the task of species classification from a database of photos taken from wildlife camera traps [5, 31]. The dataset is unbalanced, with most images containing no animal, and the distribution of camera locations and species changes between the in domain (ID) and out-of-domain (OOD) subsets.

**Amazon-WILDS** This dataset considers the task of predicting the number of stars associated with the text of a given Amazon review [43, 31]. The reviewers are different in the training set versus the test set, and the task is to perform as well as possible on this set of new reviewers. In addition to the number of stars, we also consider model performance stratified by different product types, which highlights minority subgroups whose categorization is not visible to the model.

# 4 Experimental Setup

## 4.1 Models and Training

**Vision** For computer vision datasets, we finetune BiT [32], a recently-proposed family of vision models which have achieved state-of-the-art performance on several vision tasks. We primarily consider the BiT-M-R50x1 model, pretrained on ImageNet-21k [13]. To explore the effectiveness of larger architectures and pretraining sources, in Section 6.2 we also consider performance achieved by the same-size BiT-S-R50x1, trained on ImageNet-1k, and the deeper BiT-M-R101x1 model, also trained on ImageNet-21k. These models have been shown to have emergent few-shot learning abilities, where strong classifiers for new tasks can be obtained by simply finetuning on tens or hundreds of examples with typical gradient descent techniques (rather than meta-learning techniques, for example).

**Text** For the text dataset (Amazon), we use RoBERTa-Large [37], another pretrained model with similar properties as BiT, and a representative of the BERT [15] family of models which together have obtained state-of-the-art scores on modern NLP benchmarks [60].

Other details, including hyperparameters and seed set/acquisition sizes are deferred to Appendix B.

**Random acquisition baseline** As a running baseline, we compare to the same model finetuned with a *random acquisition* function (equivalent to not doing AL). That is, $a(x; \mathcal{M}) = \text{rand}(0, 1)$, so we simply sample a random batch of data from the pool at each acquisition step.

**Comparison with non-pretrained models** To examine whether effective AL is a result of the pretraining process, we also compare to the performance observed when applying AL to a *randomly initialized*, instead of pretrained, BiT-M-R50x1.

# 5 Results

## 5.1 Accuracy per acquisition

For a general measure of success, in Figure 2 we plot the accuracy of AL versus random sampling on the validation datasets as a function of the number of samples acquired during training.

**Waterbirds** Waterbirds is evaluated on a balanced dataset where the foreground and background are not correlated. In this setting, uncertainty sampling achieves a **+11% improvement** in average validation accuracy over random sampling (Figure 2a). This comes primarily from a **+25% average increase** across the landbird-on-water and waterbird-on-land images (i.e. those without the spurious

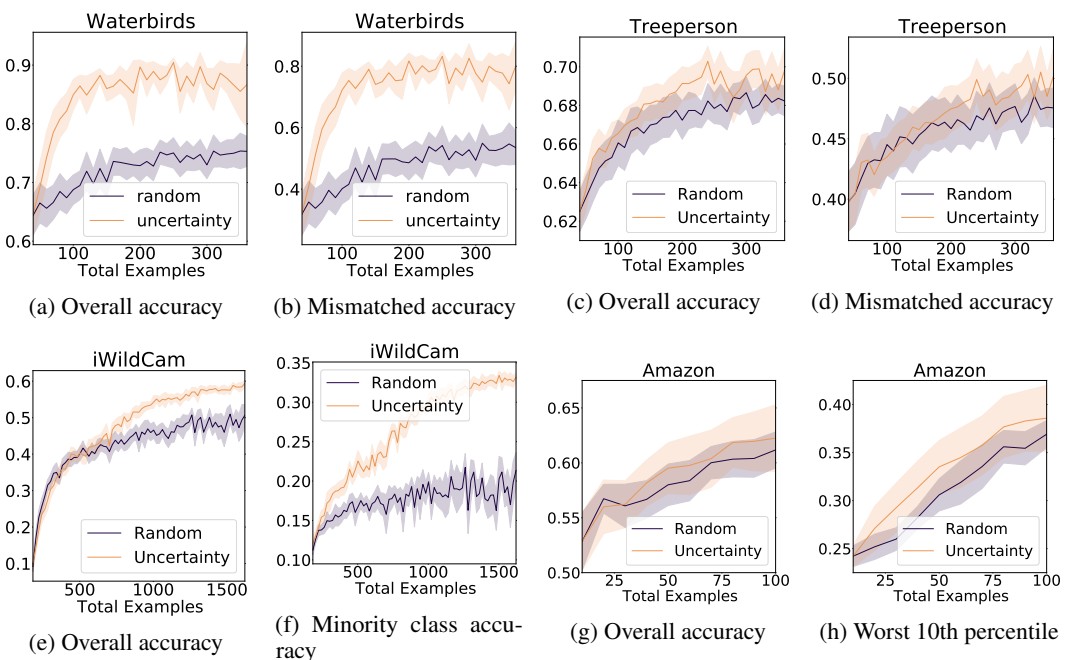

(a) Overall accuracy

(b) Mismatched accuracy

(c) Overall accuracy

(d) Mismatched accuracy

(e) Overall accuracy

(f) Minority class accuracy

(g) Overall accuracy

(h) Worst 10th percentile

Figure 2: **Uncertainty sampling outperforms random sampling on all datasets, especially on minority classes.** Class-balanced accuracies displayed for Figure 2f. Shaded regions represent 95% CIs (Gaussian approx.).

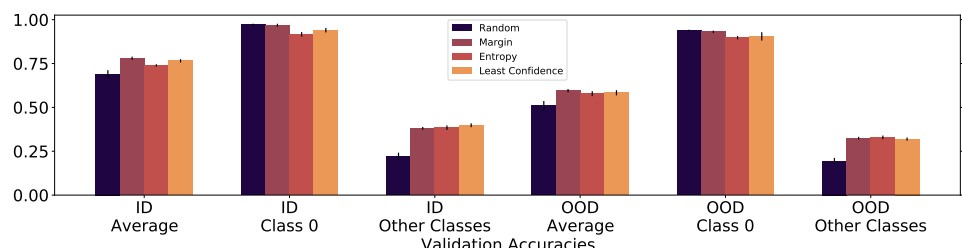

Figure 3: **All types of uncertainty sampling outperform random sampling on iWildCam.** Class 0 represents the majority class in iWildCam (no animal present).

correlation; Figure 2b). Uncertainty sampling required **5x fewer labels** than random sampling to achieve random sampling's final accuracy.

**Amazon**   In the Amazon dataset, we also see gains from AL, including **+1% on average** across reviewers, and **+2.5% on the worst 10th percentile** (Figure 2g). This suggests that our AL recipe may be of use outside of BiT or vision settings more broadly. While the final difference between uncertainty and random sampling is not large, it is statistically significant. Uncertainty sampling required **1.3x fewer labels** than random sampling to achieve the same final accuracy. This smaller effect size may also be due to the less dramatic distribution shift in the Amazon dataset. It is perhaps noteworthy that AL still succeeds in such a setting.

**iWildCam**   With the iWildCam dataset, uncertainty sampling achieved a **+9% improvement** upon random sampling. Uncertainty sampling also required **1.8x fewer labels** than random sampling to achieve random sampling's final accuracy (Figure 2e).

**Treeperson**   In the Treeperson dataset, uncertainty sampling is **+2% improved** over random sampling by the end of training (Figure 2c). Uncertainty sampling required **1.6x fewer labels** than random sampling to achieve random sampling's final accuracy.

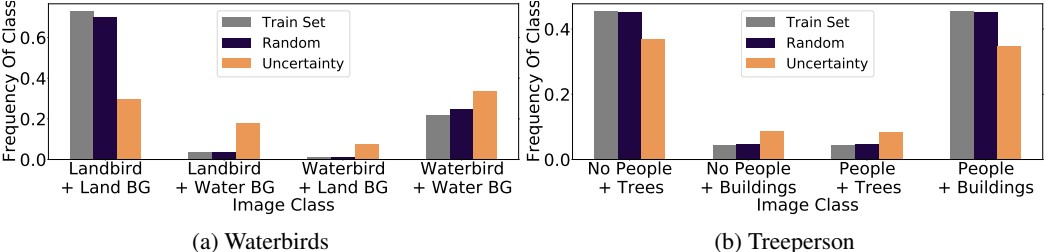

|       |       |
|-------|-------|
| (a) Waterbirds | (b) Treeperson |

Figure 4: **Uncertainty sampling identifies and upsamples disambiguating examples.** For both Waterbirds and Treeperson, uncertainty sampling selectively acquires examples where the spurious and core features disagree. Y-axis: frequency of class in acquisitions. Oversampling is visible for subgroups where uncertainty sampling acquires examples above random chance.

## 5.2 Additional AL methods

We consider two additional AL methods in addition to least confidence sampling: 1) entropy sampling, which chooses the example that maximizes the entropy of the model's predictive distribution, and 2) margin sampling, which chooses the example with the smallest difference between the first and second most probable classes [51, 52]. We run experiments with all methods on the 182-class iWildCam dataset.[2] All methods significantly outperform random sampling (Figure 3). Furthermore, margin sampling appears to slightly outperform the other two AL methods we consider, suggesting that it may a superior AL approach in multiclass settings.[3]

## 6 Analysis

### 6.1 AL with pretrained models selects examples that resolve task ambiguity

Overall, we attribute improved performance to pretrained models' ability to identify and preferentially sample examples that resolve task ambiguity in the data. For example, for Waterbirds and Treeperson, we see the model select examples with atypical background combinations, as one might intuitively hope. Similarly, for Amazon and iWildCam we see the model upsample rare types of examples, even when they are not explicitly marked in the data.

**Waterbirds** Figure 4a depicts the rate at which uncertainty sampling acquires examples of each subgroup compared to the expected rate at which random sampling would acquire examples from those same subgroup. Examples where the bird and background are mismatched are heavily oversampled. We emphasize that these minority examples are not simply members of the minority *class* (waterbirds). Instead, the model identifies and preferentially upsamples *disambiguating* examples where the spurious feature (background) and the causal feature (bird type) disagree.

**Treeperson** For Treeperson we see the same pattern as in Waterbirds: the model identifies and upsamples examples where only one of the spurious or causal features is present, despite the spurious feature being latent (Figure 4b).

**Amazon** We also see similar behavior in the Amazon dataset, indicating our method's applicability to multiple modalities and pretrained models. Not only does the model upsample lower star ratings, which are less common, it can also upsample rarer product categories—a latent attribute (Figure 5).

### 6.2 Pretraining is the key ingredient in our experiments

What drives the success of AL in our experiments? We hypothesize that better AL is an *emergent property of the pretraining process*, and evaluate this hypothesis by comparing pretrained models

---

[2]Note that least confidence, entropy, and margin sampling are identical in the case of binary classification.

[3]Intuitively, it makes sense that margin sampling would result in higher accuracy, as it chooses examples based on their decision-relevant uncertainty: the difference between the predicted class and the most likely alternative.

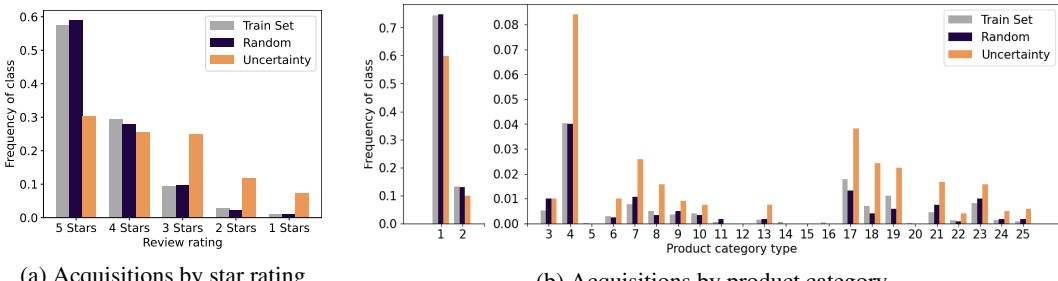

(a) Acquisitions by star rating

(b) Acquisitions by product category

Figure 5: **Uncertainty sampling upsamples both visible and latent minority subgroups.** Fraction of Amazon examples acquired by random and uncertainty sampling, stratified by star rating and product category. Upsampling is visible when the bar for uncertainty sampling is greater than the base prevalance in the unlabeled dataset available during training. Uncertainty sampling preferentially acquires examples with lower star ratings and rarer product categories, despite the latter attribute not being visible to the model. Note the separate y-axis for product categories 1 and 2 in (b).

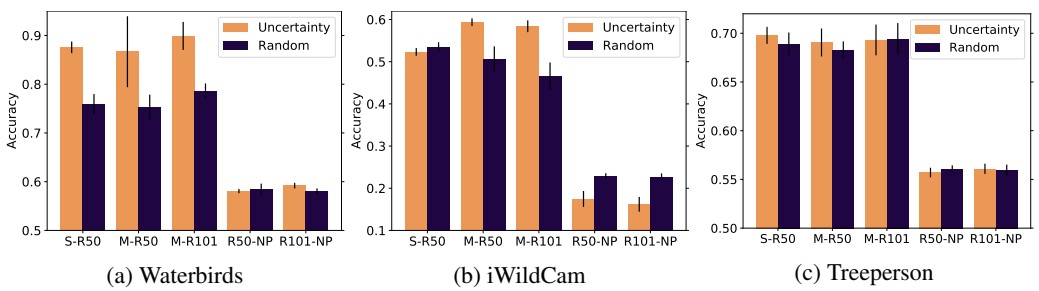

(a) Waterbirds

(b) iWildCam

(c) Treeperson

Figure 6: **Uncertainty sampling only provides gains when using pretrained models.** S-R50, M-R50, and M-R101 correspond to the BiT-S-R50x1, BiT-M-R50x1, and BiT-M-R101x1 pretrained models, respectively, while R50-NP and R101NP correspond to ResNet models which are not pretrained. Error bars represent 95% CIs (Gaussian approx.).

to their corresponding non-pretrained counterparts. We also examine the effect of *model scale* on success at AL: if a model has been trained on more data (and has presumably learned to extract more semantically-relevant features), does this enable more effective AL?

Concretely, we conduct AL experiments with 3 pretrained models: BiT-S-R50x1, BiT-M-R50x1, BiT-M-R101x1, and their corresponding non-pretrained versions. We find that for our experiments with pretrained models, uncertainty acquisition outperformed random acquisition (Figures 6a, 6b and 6c). Importantly, AL on the non-pretrained models provided no or even negative benefit, even after exploring a range of different hyperparameter configurations. These controlled experiments provide strong evidence that pretraining is indeed crucial in our setting. That said, we do not claim that pretraining is the *only* way to enable good AL in settings of task ambiguity; other methods of addressing the so-called "cold start" problem (e.g. [20, 63]) may also prove fruitful or complementary.

**Effect of scale** In one case, we also see a demonstrable effect of *scale* on the efficacy of the AL process: the BiT-S-R50x1 model, which was pretrained on a smaller dataset than the BiT-M models (ImageNet-1k vs ImageNet-21k) fails to outperform random sampling on iWildCam, in contrast to the two other models pretrained on more data. This suggests a potential scaling trend for AL, where gains from AL may continue to grow as pretrained models are trained for longer on more data. However, we did not see a difference between BiT-M-50x1 and BiT-M-101x1, which were trained on the same dataset but have different numbers of parameters. This may be because dataset size must be increased jointly with parameter count to see continued gains from scaling [29, 26].

**Impact of pretraining on acquisition patterns** As an additional cross-check, we also observe that pretrained models acquire disambiguating subgroups much more efficiently than their non-pretrained counterparts. See Appendix G for additional figures and results.

| | No Finetuning | Finetune On Seed Set (40) | Finetune On Seed Set (40) + 20 |
|---|---|---|---|
| Average | 0.402 | 0.42 | 0.424 |
| Landbird /LandBG | 0.389 | 0.416 | 0.42 |
| Waterbird /LandBG | 0.63 | 0.653 | 0.733 |
| Landbird /WaterBG | 0.426 | 0.467 | 0.447 |
| Waterbird /WaterBG | 0.435 | 0.418 | 0.419 |

| | No Finetuning | Finetune On Seed Set (40) | Finetune On Seed Set (40) + 20 |
|---|---|---|---|
| Average | 0.311 | 0.32 | 0.34 |
| Landbird /LandBG | 0.306 | 0.321 | 0.369 |
| Waterbird /LandBG | 0.194 | 0.316 | 0.325 |
| Landbird /WaterBG | 0.286 | 0.248 | 0.262 |
| Waterbird /WaterBG | 0.337 | 0.33 | 0.255 |

(a) Group accuracies for linear classifier on Waterbirds image embeddings attained from a pretrained BiT model after various degrees of finetuning

(b) Group accuracies for linear classifier on Waterbirds image embeddings attained from an non-pretrained BiT model after various degrees of finetuning

Figure 7: **Both causal and spurious features are more linearly separable in pretrained models.**

## 6.3 Pretraining yields a better feature space for AL

While pretraining clearly improves the AL process, the mechanisms behind this improvement remain unclear. Given the strong theoretical results AL enjoys in the linear setting [2, 3, 41], we hypothesize that pretraining may aid AL by *linearizing* the features salient for task ambiguity. This hypothesis is further inspired by recent studies finding that a wide range of features are linearly separable in the feature spaces of large pretrained models [10].

To quantify this, we train linear classifiers on the second to last layer of the BiT models. The classifiers are trained to predict each image's bird type and background type (4 classes total, rebalanced to comprise 25% of the data). As shown in Figure 7, these classes are indeed far more linearly separable in pretrained models, providing evidence for this hypothesis.

We present additional investigations of t-SNE plots for pretrained and non-pretrained models in Appendix H, which demonstrates increased separation of latent classes for pretrained models, as well as how the acquired examples are closer to the class boundaries.

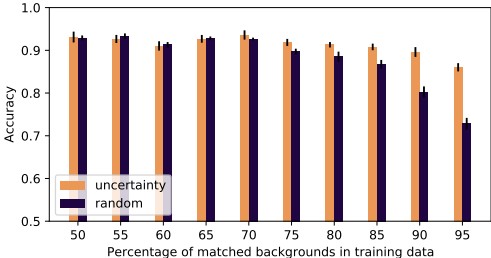

Figure 8: **Task ambiguity is the key factor driving the success of AL with pretrained models.** Accuracy on Waterbirds out-of-domain validation set for pretrained BiT-M models finetuned on datasets with different fractions of matched backgrounds. As disambiguating examples become more scarce, uncertainty sampling experiences far less of an accuracy drop vs random sampling.

## 6.4 How does the degree of task ambiguity impact AL?

Finally, we measure the impact of the *strength* of task ambiguity on AL. To do so, we construct variants of the Waterbirds dataset where the percentage of mismatched examples range from 95% to 50% but the marginal class probabilities remain fixed.[4] We then proceed with AL and report results in Section 6.3. We observe a clear trend where the gains from AL gradually increase as the percentage of mismatched examples increases to 95%. We find similarly clear trends in the upsampling ratio of mismatched backgrounds, shown in Figure 9. These results provides further evidence that task ambiguity is the key driving factor behind the success of AL in this setting.

## 6.5 Failure Cases

We also encounter some failure cases when training on datasets far from the distribution of the pretrained model. We perform preliminary experiments on Camelyon17-WILDS [4, 31], which considers tumor identification from tissue patches, and FMoW-WILDS [11, 31], which considers land-use classification from satellite images. AL performs comparably or worse than random sampling on these datasets, even when using a pretrained BiT model, suggesting that specialized models may be necessary to see gains in domains very different from ImageNet-21k.[5]

---

[4]We construct these datasets using the code at `https://github.com/kohpangwei/group_DRO`.

[5]However, we note that Camelyon17-WILDS is also known to exhibit high variance across seeds (see `https://wilds.stanford.edu/get_started/`) which may also be a contributing factor.

# 7 Related work

**Task ambiguity and specification** Several works address ambiguity or poor specification in machine learning problems. [58] describe the problem of "inductive ambiguity identification," and describe AL as a promising potential solution that has failed to see practical success. [12] describe the problem of *underspecification*, where high variance, instability, and poor model performance result from training overparameterized models that are underconstrained by their training datasets. [21] describes how task ambiguity can arise when both desirable and undesirable features are predictive of the training labels, a problem which several works seek to better characterize and address [42, 49, 55, 50]. Finally, [18] address task ambiguity in few-shot settings via a probabilistic meta-learning algorithm, and perform an AL experiment in a 1D regression setting. We build on these works by demonstrating that simple uncertainty sampling with pretrained models can be an effective approach to the task ambiguity problem across a wide variety of high-dimensional classification settings—including when the sources of task ambiguity are not known.

**Uncertainty and distribution shift** In the face of these challenges, several works have tried to quantify how much pretrained models know about problems or their own uncertainty about them. [47] propose a question answering dataset with unanswerable questions, where a model must abstain rather than proceeding with an answer. Pretraining can also improve the calibration of model uncertainty [24] and pretrained features can be used for out-of-distribution detection [48, 62]—observations that align with our findings that uncertainty sampling can identify minority subgroups in datasets. A related stream of work seeks to identify high-confidence examples that are predicted incorrectly [1, 34]; by contrast, our focus is on improving model behavior across the full range of examples. Finally, our observation that upsampling latent minority groups results in better performance aligns well with [50, 28, 36], which explore various upweighting or upsampling strategies. Importantly, however, our approach does not require these groups to be known in advance.

**AL and example selection** Active learning (AL) [35, 53, 52, 27] is a well-studied field that investigates how machine learning algorithms might automatically select helpful additional data points to maximize their performance. Such strategies are especially helpful in imbalanced settings [17, 40] and have been fruitfully applied to deep models [19, 6], including pretrained models [63, 39, 54]. Past work has also considered AL for few-shot learning [61]. We extend these works by considering AL for resolving task ambiguity, showing that pretrained models successfully choose examples based on their high-level semantics, such as atypical backgrounds or rare latent attributes. Also in contrast to prior work, we investigate the *role of pretraining itself* by performing equivalent experiments with non-pretrained models, and providing a potential mechanism for the difference.

**Pretrained models and their emergent properties** Our work contributes to a broader literature on how pretraining enables new kinds of model capabilities [7, 56], especially those holding across multiple domains [57]. For example, [8] identify the phenomenon of in-context learning, where tasks can be specified for models through a language modeling prompt, while [9] discover that a self-supervised vision model implicitly learns high-quality segmentation maps visible through attention scores. [29, 25] conduct scaling laws experiments which chart how capabilities emerge with scale. We identify a new model capability that is significantly improved by pretraining: the capacity to actively learn and resolve task ambiguity in high-dimensional settings.

# 8 Discussion and Limitations

We show that pretrained models can preemptively resolve task ambiguity through active learning (AL), without requiring humans to anticipate these possible failure modes in advance. We find that AL helps across a variety of settings where data is spuriously correlated, undergoes domain shift, or contains unlabeled subgroups. These behaviors emerge most clearly as a result of large-scale pretraining, suggesting that AL may be an underappreciated tool for increasing the reliability of systems in real-world settings.

Of course, AL is no cure-all for resolving task ambiguity. First, it requires a human in the loop, which increases the time required to train a model compared to random sampling. Second, it requires the labeling method to be relatively free of noise—this may be acceptable if annotators are domain experts or are well-trained, but may also increase the cost per acquired example. Third, it is limited

by the range of examples present in the unlabeled dataset—a model cannot elicit labels for examples that do not exist.

Finally, we note the opportunity for an exciting array of future work, including broader investigation of these methods across domains such as medical, scientific, or industrial settings, as well as better understanding how pretraining shapes AL as models continue to scale.

## Acknowledgments and Disclosure of Funding

We would like to thank Shyamal Buch, Shreya Shankar, Megha Srivastava, and Ethan Perez for helpful comments. AT is supported by an Open Phil AI Fellowship.

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
