# A   Code release

Code and training scripts are available at:
`https://github.com/alextamkin/active-learning-pretrained-models`

# B   Additional experimental details

Here we provide some additional experimental details.

All BiT runs use the same default settings specified in the BiT paper [32] and the BiT GitHub repo:
`https://github.com/google-research/big_transfer`

Settings that are constant across all BiT runs:

1. **Optimizer** - SGD with momentum 0.9
2. **Learning Rate** - Base learning rate 0.003. Linear warm up to this rate, then staircase decay. Exact schedule depends on dataset size, but for our few shot setting, this means: (1) linear warm up in the first 100 steps to 0.003, then (2) decay 10 fold every 100 steps; (3) after 500 steps, stop and move on to the next acquisition round.
3. **Data Augmentation** - Random cropping and flipping. See our `repo/utils/datasets/load` for details. Also available in BiT repo.
4. **Batch Size** - 32 when training, split into gradient accumulation microbatches of size 8.
5. **Early stopping condition** - When the training loss reaches below 0.001 times the original training loss.

Settings that varied between image datasets:

1. **Size of Initial Seed Set** - 40 for Waterbirds, 40 for Treeperson, 182 for iWildCam
2. **Size of Training Pool** - Entire training set for Waterbirds, Treeperson. A random subset of 12000 examples (re-drawn for each acquisition) from the entire training set of iWildCam.
3. **Number of Acquired Examples** - 320 for Waterbirds, 320 for Treeperson, 1456 for iWildCam.
4. **Number of Examples Acquired Each Acquisition** - 5 for Waterbirds, 5 for Treeperson, 20 for iWildCam.

Settings used for Amazon + RoBERTa-Large runs:

1. **Optimizer** - AdamW with default hyperparameters ($\beta_1 = 0.99, \beta_2 = 0.999$, weight decay = 0.1).
2. **Learning Rate** - LR = 1e-6.
3. **Batch Size** - 2 (due to memory considerations).
4. **Early Stopping Condition** - When the training loss reaches below 0.001 times the original training loss.
5. **Size of Initial Seed Set** - 10.
6. **Size of Training Pool** - A random subset of 2000 examples (re-drawn for each acquisition) from the entire training set.
7. **Number of Acquired Examples** - 90.
8. **Number of Examples Acquired Each Acquisition** - 2.

# C   Treeperson dataset

The Treeperson dataset is composed of images from Visual Genome [33] with different compositions of detected objects.

Training set composition by subclass:

- Person and Building: 3700
- Person and Tree: 370
- No Person and Building: 370
- No Person and Tree: 3700

The validation set contains 498 examples of each subclass.

The following annotated objects were used to form the different subclasses:

- Person: person, people, man, men, woman, women
- Building: building, buildings
- Tree: tree, trees, leaf, leaves, grass

The training set and validation set were drawn randomly from qualifying images in Visual Genome's training set and validation set, respectively.

## D  Early stopping condition

At the outset of this work, we explored if we could identify a heuristic for stopping training when there was no validation set present. We compared how the BiT model would perform if it stopped the finetuning step based off of when the validation accuracy plateaued versus when the training loss decayed to be 0.1% of its original value.

We ran a smaller experiment than the standard Waterbirds parameters we described in Appendix B. Namely, our seed set was of size 32, we acquired 64 examples on top of that, and we acquired one example at a time. We also defined the validation accuracy as having plateaued the fifth time it did not increase.

We ran twelve paired experiments where for twelve different randomized seed sets, we performed the Waterbirds experiment four times - with random sampling and stopping when validation accuracy plateaued, with random sampling and stopping when training loss decayed, with uncertainty sampling and stopping when validation accuracy plateaued, and with uncertainty sampling and stopping when training loss decayed.

We found that these experiments achieved:

1. Random sampling + Stop from validation accuracy: average accuracy = 69.11%, average duration =  3.5 hours
2. Random sampling + Stop from training loss: average accuracy = 69.93%, average duration = 38 minutes
3. Uncertainty sampling + Stop from validation accuracy: average accuracy = 83.64%, average duration =  4.5 hours
4. Uncertainty sampling + Stop from training loss:  average accuracy = 85.62%, average duration = 74 minutes

Thus, we concluded that the by stopping our finetuning step just by waiting for the training loss to decay to 0.001 of its original values, we could achieve comparable if not better accuracies, spend a fraction of the time, and remove the need for a labeled validation set.

We did not consider other values besides 0.001 when conducting this search, though one could consider optimize this number in a fair way on a separate set of labeled datasets (in general one should not optimize AL pipelines using the validation sets they are being evaluated on).

## E  How the degree of task ambiguity impacts acquisitions

In Figure 9, we also plot the degree of upsampling that occurs for mismatched examples for different degrees of task ambiguity. As disambiguating examples become more scarce, uncertainty sampling increasingly upsamples them relative to their prevalence in the unlabeled training set.

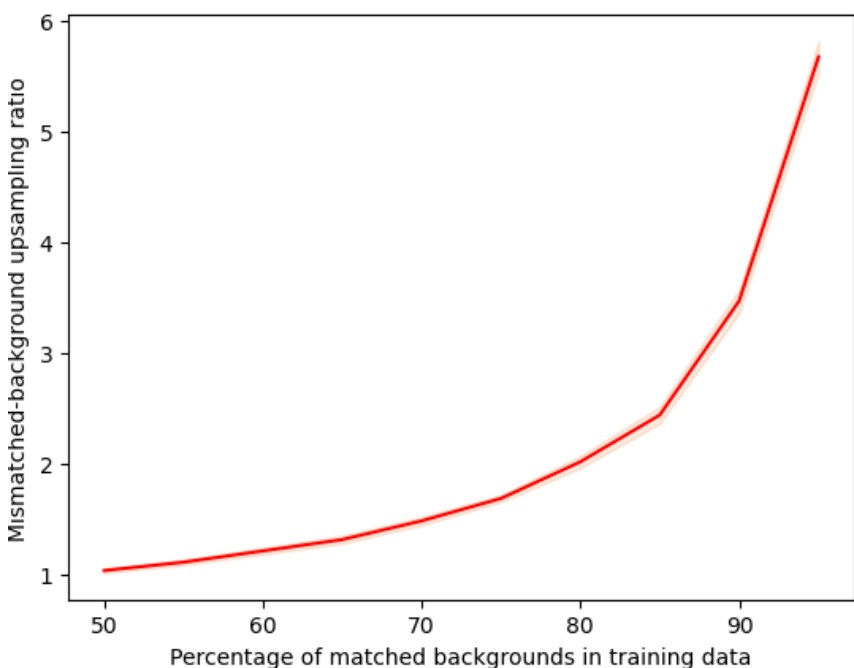

Figure 9: **Stronger task ambiguity results in more upsampling of disambiguating examples**
Upsampling ratio of Waterbirds examples with mismatched backgrounds for pretrained BiT-M models
finetuned on datasets with different fractions of matched backgrounds. As disambiguating examples
become more scarce, uncertainty sampling increasingly upsamples them relative to their prevalence
in the unlabeled training set.

## F   Vision transformer model

We broaden our coverage of computer vision models to include vision transformers [16], the other
major architecture family currently in use. We train ViT-16/B,[6] on Treeperson. (Figure 10). This
vision transformer was pretrained on ImageNet-21k for fewer epochs then BiT (9 vs 70), which is
reflected by its lower oversampling of minority classes and comparatively smaller gains in accuracy
compared to BiT (Figure 11).

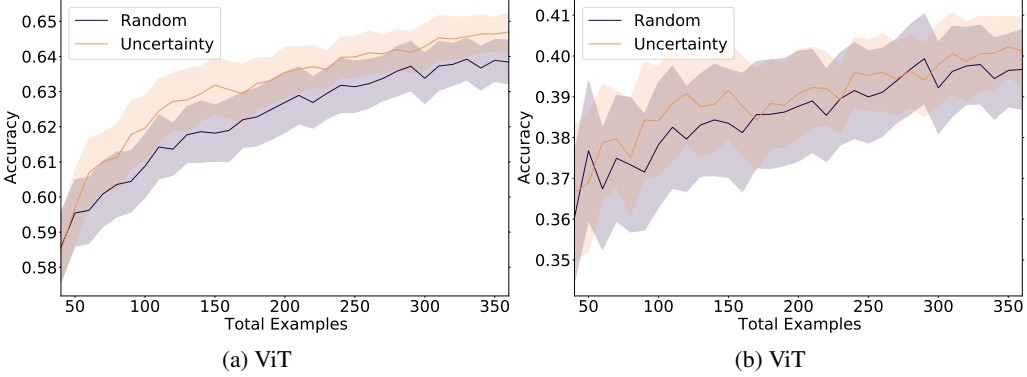

(a) ViT                                         (b) ViT

Figure 10: **AL improves upon random sampling for Treeperson when using ViT-B/16.** However,
the gains are not as large as for BiT, which was pretrained for 8x more epochs.

---

[6]https://huggingface.co/google/vit-base-patch16-224

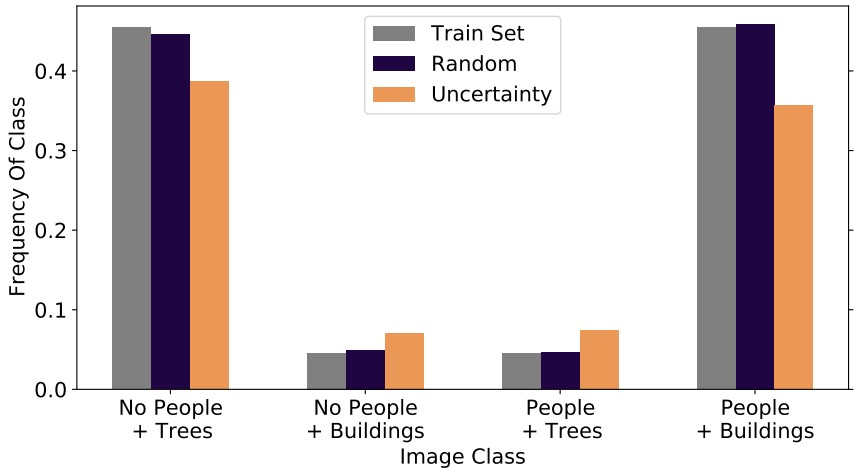

Figure 11: **Acquisitions for Treeperson using ViT.** The ViT model requested labels for minority classes at a significantly lower rate than did the BiT model. However, it still upsamples atypical backgrounds more than random sampling.

## G   Effect of model scaling and pretraining on acquisition

For the Waterbirds model scaling experiment, we track the examples each model acquires. These are presented in Figure 12. The acquisition patterns of all the pretrained models look fairly similar—they upsample both minority (landbird/water-background, waterbird/land-background) subclasses. However, the non-pretrained model is unable to capture that distinction, and is only able to upsample images with a water background, resulting in worse performance. A summary of final class acquisition ratios is available in Figure 13.

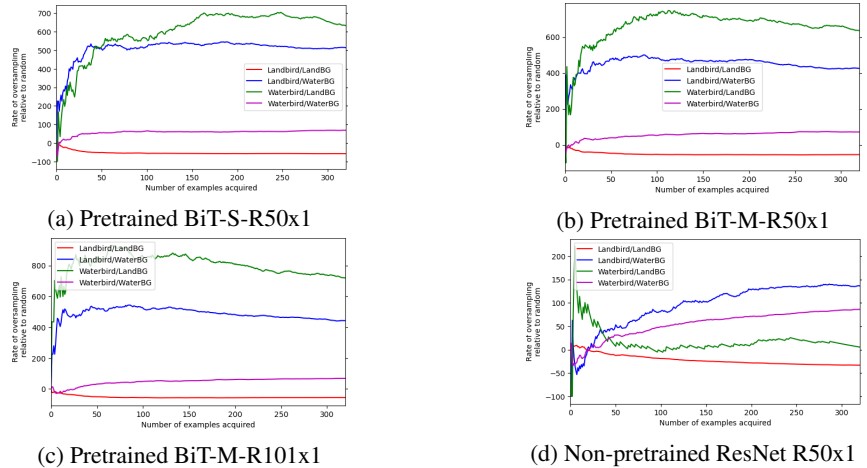

(a) Pretrained BiT-S-R50x1

(b) Pretrained BiT-M-R50x1

(c) Pretrained BiT-M-R101x1

(d) Non-pretrained ResNet R50x1

Figure 12: **All pretrained models acquire disambiguating subgroups much more efficiently than their non-pretrained counterparts on Waterbirds.** The pretrained models do not simply oversample based on the the bird or the background; instead they oversample disambiguating examples which have mismatched backgrounds. By contrast, the non-pretrained model only oversamples images with a water background, and accordingly is less able to perform well on the balanced validation set.

## H   Visualization Of Image Embeddings Of Pretrained Bit-M

**t-SNE visualization**    We visualize the second-to-last layer embeddings of BiT (without any finetuning) using t-SNE [59].

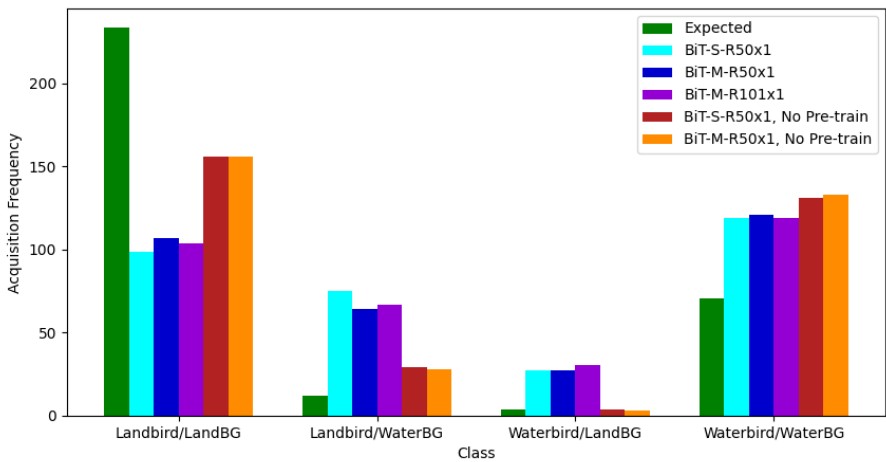

Figure 13: **Pretrained models ask for labels of images with mismatched backgrounds, while non-pretrained models do not.**

As seen in Figure 14b, when performing t-SNE on the Waterbirds image embeddings, we find that the t-SNE of the pretrained model has much more structure than that of the non-pretrained model. In the non-pretrained model, the image embeddings of the landbird/land background, landbird/water background, and waterbird/water background classes are distributed uniformly about the center of the t-SNE projection. However, we observe much more structure in the pretrained model's t-SNE. In particular, the most noticible difference is that the landbird/water background and waterbird/land background classes can be found near the other landbird and waterbird images, respectively. This shows that even without any finetuning, the pretrained model already has learned features to help process the type of bird that appears in the image.

In the t-SNE plots, the black stars represent the embeddings of the first 10 images that were acquired by each model. For the pretrained model, the acquired examples very clearly fall along the waterbird-landbird boundary in the projected feature space. However, in the non-pretrained model, the acquired examples are distributed randomly. This suggests that the features acquired during pretraining enables them to select examples that fall near the decision boundary of a new unseen task.

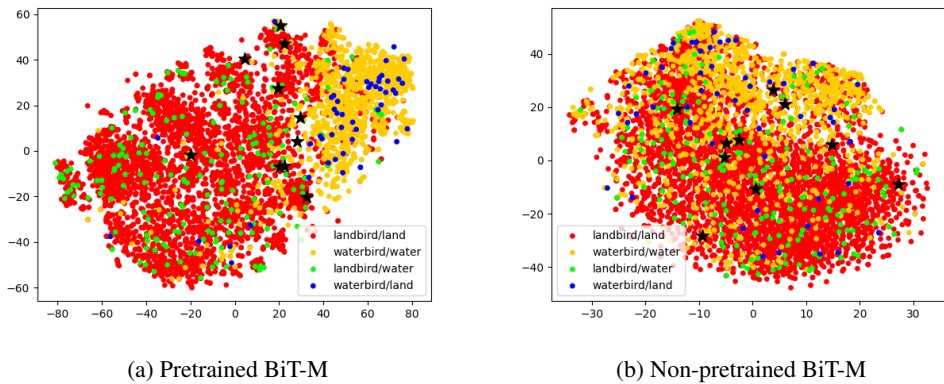

(a) Pretrained BiT-M             (b) Non-pretrained BiT-M

Figure 14: **Without any finetuning, pretrained models already embed images into a useful feature space.** t-SNE of Waterbirds image embeddings from pretrained and non-pretrained BiT-M. Black stars denote acquired examples, which fall more consistently along the decision boundary between landbirds and waterbirds for pretrained models vs non-pretrained ones.

**Potential Societal Impact** Our work concerns more general conceptual understanding and use of pretrained models, making it hard to speculate directly about downstream societal impacts. In general, such methods are likely to amplify the existing uses of such models. For example, methods for actively aligning models to human intent could have positive or negative consequences depending on whether the intent was beneficent or malevolent. This highlights the need for careful design of societally-beneficial governance structures when deploying these models.

**Compute used** We used Titan Xp GPUs in an internal cluster. Each trial took on the order of hours.