# OpenReview forum: "Active Learning Helps Pretrained Models Learn the Intended Task"
_NeurIPS.cc/2022/Conference — NeurIPS 2022 Accept_

### Official Review · Reviewer_GGwr · 2022-06-13

**Rating:** 7
**Confidence:** 3
**Soundness:** 4 excellent
**Presentation:** 4 excellent
**Contribution:** 2 fair

**Summary:**

The authors show that under active learning, pre-trained models benefit from fine-tuning on samples where they are less confident. In contrast, models trained from scratch perform similarly or worse when trained in this way - simply sampling random batches is optimal. They show that this performance gain is modulated by how ambiguous the task is, such as the presence of confounders. In particular, a pre-trained model learns a feature space in which the causal and confounding attributes are more linearly separable, suggesting that it can more easily clarify the intended task by identifying disambiguating instances.

**Questions:**

Would be helpful to briefly discuss when the least confidence heuristic might be more or less appropriate than entropy and margin sampling. Even if their empirical performance is similar, can you think of reasons to choose one over another?
It seems that there should be some way to extend or connect these findings outside of active learning, e.g. in curriculum learning, continual learning, or even in standard learning settings with distributional shift. Something to think about.

**Limitations:**

Limitations are well characterized.

**Strengths And Weaknesses:**

Strengths
The paper is clearly written, and presents clear, thorough experiments to demonstrate and explore the behavior of pre-trained models in both vision and language tasks. The hypothesis is clear, and Fig. 1 is really nice. There are interesting findings - the pre-training dataset size matters more than model size, and the benefit of the least confidence heuristic disappears if there are no explicit confounders!
The figures and text are all very clear.

Weaknesses
I feel the findings are actually quite pessimistic - the least confidence heuristic requires quite a few conditions to hold in order to be effective (task ambiguity with distribution shift, and a model pre-trained on a sufficiently large dataset). So what can be done when the conditions don't hold? Is random labeling really the best option in other cases?
The overall scope and contributions seem a little narrow. It comes off as a careful examination of a very specific phenomenon.
The paper only performs experiments on classification, where active learning is less useful since labels tend to be much cheaper than e.g. video or volumetric segmentation.

---

> ### Author Response · Authors · 2022-08-02
> **Response**
>
> We thank the reviewer very much for their review! We are happy they thought our paper presented "interesting findings," was "clearly written, and presents clear, thorough experiments."
>
> **"The least confidence heuristic requires quite a few conditions to hold in order to be effective (task ambiguity with distribution shift, and a model pre-trained on a sufficiently large dataset)."**
>
> We actually believe these conditions are very common—in fact we think they might even be the default setting when using large pretrained models!
>
> Specifically, pretrained models are becoming increasingly common, and make it easy to train a classifier with just a few examples—this is exactly the setting where models are likely to be confused between multiple discriminative features. Furthermore, real-world applications of ML almost always experience distribution shifts through data drift or other shifts over time [1, 2].
>
> This means we could see active learning potentially becoming a standard way to train large pretrained models in the future.
>
> [1] WILDS: A Benchmark of in-the-Wild Distribution Shifts, Koh et al, 2021
> [2] Rethinking Streaming Machine Learning Evaluation, Shankar et al, 2022
>
> **"The paper only performs experiments on classification, where active learning is less useful since labels tend to be much cheaper than e.g. video or volumetric segmentation."**
>
> Annotation cost is certainly important to consider, but complex outputs are not required for high annotation cost. Many classification tasks have very high annotation cost: for example, the iWildCam dataset, which we study, includes camera trap photos of 206 different species, requiring expert biologists from the National Park Service and USGS to provide labels. Many other important domains (e.g. manufacturing, biology, legal document understanding, medical imaging) also have high annotation cost, either because they require expensive domain expertise (e.g. radiologists) or because they require costly experiments (e.g. microbiological assays). These concerns have long motivated classification as the main setting for AL [3]. However, we certainly agree that video or volumetric segmentation would be interesting extensions to our work.
>
> [3] Active Learning, Settles, 2012
>
> **Least confidence vs margin sampling vs entropy**
>
> For binary classification tasks, these methods are all equivalent. For multi-class classification it appears that margin sampling actually outperforms least confidence, so we'd be inclined to recommend it as the default choice. Intuitively, margin sampling is useful because the margin is the portion of the model's uncertainty that is directly relevant for the network's decision (i.e. that would result in the model making a different choice). We'll add this discussion to the paper.
>
> **Connections to curriculum learning, continual learning, or distributional shift**
>
> We completely agree! One natural connection is that active learning is one strategy for enabling a model to select its own curriculum (though certainly not the only way).
>
> The connection between distributional shifts, which we study in our paper, and continual learning is also interesting. One form of continual learning involves learning to perform the same task on a new domain. If the model gets access during training to unlabeled examples from this domain, active learning could help the model adapt more quickly, while ensuring it retains its previous capabilities. This is similar to the setting explored in [4].
>
> [4] Domain Adaptation meets Active Learning, Rai et al, 2010

---

> > ### Comment · Reviewer_GGwr · 2022-08-04
> > **Response**
> >
> > I think the authors' response is reasonable. However I still feel some part of the takeaway is unclear. To me the point of this paper is to say "uncertainty sampling AL works much better on pretrained models and here is why." Then I have 2 questions:
> > 1. Uncertainty-based AL has apparently done well without pretrained models in previous works. Why?
> > 2. If we shouldn't use uncertainty sampling for non-pretrained models, what should we use instead?

---

> > > ### Author Response · Authors · 2022-08-08
> > > **Response**
> > >
> > > We thank the reviewer for their response! These are great questions:
> > >
> > > **"Uncertainty-based AL has apparently done well without pretrained models in previous works. Why?"**
> > >
> > > We certainly do not intend to dismiss a long line of work applying AL to different problems. These successes are best established in the linear setting, where a robust theoretical backbone characterizes the settings where AL algorithms can succeed and to what degree we can expect improvements from them [1,2,3].
> > >
> > > However, in more complex, nonlinear settings, these guarantees do not hold. While some researchers have reported successes, systematic studies across many tasks have found that no single AL method reliably improves over random sampling [4,5]. Anecdotally, we and many of our colleagues have struggled to get AL to work in practice on our problems, and we suspect there may be some degree of selection bias at play in the literature.
> > >
> > > **"If we shouldn't use uncertainty sampling for non-pretrained models, what should we use instead?"**
> > >
> > > We aren't claiming that nobody should use uncertainty sampling with randomly-initialized models—only that there are many situations where AL fails with randomly-initialized models where it succeeds (often dramatically) with pretrained ones.
> > >
> > > One intuitive reason this might be the case is that randomly-initialized models lack the representations necessary to distinguish the informative from uninformative examples. Following this intuition, one potential strategy for AL with randomly-initialized models could be to train them on far more examples, in order to develop some of these necessary representations (something like a very limited form of pretraining).
> > >
> > > This aligns with the findings of one of the most well-known AL papers, which reported that it can take many thousands of examples before AL starts to overtake random sampling [6].
> > >
> > > However, pretrained models have become ubiquitous in the past few years, and we expect that most researchers and practitioners interested in AL will simply start with a powerful pretrained model, specially for data-scarce settings where few-shot learning and AL are most useful (and training on tens of thousands of data points is infeasible)
> > >
> > >
> > > **References**
> > >
> > > [1] Maria-Florina Balcan, Andrei Z. Broder, and Tong Zhang. Margin based active learning. In COLT, 2007.
> > >
> > > [2] Maria-Florina Balcan and Philip M. Long. Active and passive learning of linear separators under log-concave distributions. In COLT,, 2013
> > >
> > > [3] Stephen Mussmann and Percy Liang. On the relationship between data efficiency and error for uncertainty sampling. In ICML, 2018.
> > >
> > > [4] Siddharth Karamcheti, Ranjay Krishna, Li Fei-Fei, and Christopher D. Manning. Mind your outliers! investigating the negative impact of outliers on active learning for visual question answering. In ACL 2021.
> > >
> > > [5] David Lowell, Zachary Chase Lipton, and Byron C. Wallace. Practical obstacles to deploying active learning. In EMNLP, 2019.
> > >
> > > [6] ​​Donggeun Yoo and In So Kweon, Learning Loss for Active Learning, In CVPR, 2019.

---

> > > > ### Comment · Reviewer_GGwr · 2022-08-08
> > > > **Thanks**
> > > >
> > > > This is a well-written and helpful response. I understand that the page limit restricts your ability to incorporate some of the clarifications or additional discussion. Assuming that you would incorporate those into a camera ready version, I am happy to raise the score from 6 to 7.

---

> > > > > ### Author Response · Authors · 2022-08-08
> > > > > **Thank you**
> > > > >
> > > > > Thanks so much for reading and engaging with our response! We would definitely incorporate these clarifications into the camera ready version.

---

### Official Review · Reviewer_KZEm · 2022-07-09

**Rating:** 4
**Confidence:** 4
**Soundness:** 3 good
**Presentation:** 3 good
**Contribution:** 2 fair

**Summary:**

This paper studies pre-trained models for active learning to address the problem of task ambiguity where limited training data cannot fully specify expected outputs for all possible inputs. The paper performs extensive experiments to evaluate uncertainty sampling approaches for active learning using different pre-trained models and datasets (in both vison and natural languages). To standardize the experiments, the paper proposes to remove validation set, leveraging training loss to terminate training and recommended hyperparameters from prior work. The considered datasets explore both combination of causal and spurious features and robustness under distribution shifts. Based on the experiments, the paper provides several findings for active learning with pre-trained models, i.e., uncertainty sampling is more better than random sampling for active learning, pre-trained models helps active learning to select atypical combinations of features or upsample raw types of examples, pre-training is much better than random initialization for active learning, and pre-training can learn better representations and encourage more informative selection for active learning.

**Questions:**

Please consider the weakness presented above.

**Limitations:**

The paper includes a section to describe potential limitations of current work, including increased costs for model training, human annotation, and limited contexts to unlabeled set.

**Strengths And Weaknesses:**

**Strength

+Active learning with pre-trained large-scale models is still less explored in the literature.

+The experiments are extensive and controlled. The results in this paper might provide useful baselines for future research in this important area.

**Weakness

-Some experiments in the paper are quite conventional and the findings seem to be expected given the demonstrated effectiveness of pre-trained models in many previous work. For instance, it is pretty popular in active learning research that random sampling cannot perform as well as heuristic-based selection (e.g., uncertainty sampling). Also, it is pretty expected that pre-trained models would perform better than randomly initialized models due to better accumulation of knowledge. Pre-trained models are also considered to learn better representations in general. Although the paper does contribute to verify such general knowledge for active learning and pre-trained models, it might be better to study specific challenges of active learning when pre-trained models are employed for modeling, e.g., increased training time of the large-scale models that might drive up waiting time of annotators, trade-offs of different active learning approaches with pre-trained models (e.g., uncertainty-based vs disagreement-based).

-The paper only considers classification tasks that might not well represent different tasks in machine learning, e.g., structured prediction tasks in computer vision and natural language processing. The annotation costs in such tasks might be more significant and it is interesting to explore whether pre-trained models can help to reduce the costs. Also, it is unclear if the proposed heuristics to remove the validation set can generalize to such tasks/datasets.

-Minor: An interesting observation in lines 214-216 is that larger pre-trained models have minimal effect for the accuracy of active learning methods. It might be helpful to further explore this issue. Can larger models be more beneficial if more data is selected for annotation in active learning?

---

> ### Author Response · Authors · 2022-08-02
> **Response**
>
> We thank the reviewer very much for their review! We are happy they appreciated our "extensive and controlled" experiments, and feel we contributed "useful baselines for future research in this important area."
>
> **"Findings seem to be expected given the demonstrated effectiveness of pre-trained models in many previous work"**
>
> What's surprising and novel about our findings is not what they have to say about AL or pretraining individually, but about how together they can make strides on challenging robustness problems.
>
> As we state in L281-288, it is indeed well-known that pretraining produces better representations, but it has not previously been shown that AL can use these representations to produce better out-of-domain generalization. Crucially, it is not obvious that this would be the case a priori—pretraining or AL might both independently help performance, but pretrained models might not be better active learners. To our knowledge, our work is the first to show how pretraining helps AL choose disambiguating examples, improving the generalization and robustness of the resulting models.
>
> This relationship is especially important given how pretrained models are now the predominant starting point for most practitioners, and the few-shot capabilities of these models make AL far more practical than it has been at any time in the past.
>
> **"The paper only considers classification tasks … annotation costs in [structured prediction] tasks might be more significant"**
>
> Annotation cost is certainly important to consider, but this statement makes a false equivalence between tasks with high annotation cost and tasks with complex outputs. Many classification tasks have very high annotation cost: for example, the iWildCam dataset, which we study, includes camera trap photos of 206 different species, requiring expert biologists from the National Park Service and USGS to provide labels. Many other important domains (e.g. manufacturing, biology, legal document understanding, medical imaging) also have high annotation cost, either because they require expensive domain expertise (e.g. radiologists) or because they require costly experiments (e.g. microbiological assays). These concerns have long motivated classification as the main setting for AL [3]. However, we agree that structured prediction certainly could be investigated within our framework, including revisiting prior work [e.g., 1,2] with more recent and larger pretrained models, especially in settings with domain shift.
>
> [1] Active learning for human pose estimation, Liu et al, 2017
> [2] Active learning for structured prediction from partially labeled data, Khodabandeh et al, 2017
> [3] Active Learning, Settles, 2012
>
>
> **"Minor: An interesting observation in lines 214-216 is that larger pre-trained models have minimal effect for the accuracy of active learning methods. It might be helpful to further explore this issue"**
>
> We also think this comparison is interesting. Recent work suggests that both pretraining data and model size can bottleneck model capabilities if they are not scaled jointly in the right proportion [4, 5]. One possibility is that the BiT-M-101x1 model does not improve over the BiT-M-50x1 model because only the model size was increased, not the training dataset. Unfortunately the BiT-L models, which were trained on a larger, private dataset have not been released, but we would be interested to run the experiments if they ever are. Another possibility is that the representations needed to support good active learning emerge with scale, but in an uneven and unpredictable manner, as has been observed across several other settings [6].
>
> [4] Scaling Laws for Neural Language Models, Kaplan et al, 2020
> [5] Training Compute-Optimal Large Language Models, Hoffman et al, 2022
> [6] Predictability and Surprise in Large Generative Models, Ganguli et al, 2022

---

### Official Review · Reviewer_DEK8 · 2022-07-14

**Rating:** 5
**Confidence:** 2
**Soundness:** 3 good
**Presentation:** 4 excellent
**Contribution:** 2 fair

**Summary:**

This paper investigates active learning with pretrained models to achieve better performance when using uncertainty sampling. Empirical results show that active learning with pretrained models and uncertainty sampling outperform random sampling and un-pretrained models.

**Questions:**

1. From the Figure 2, the image results is much better than the text results. Is it necessary to conduct more experiment on more text datasets?

**Limitations:**

yes

**Strengths And Weaknesses:**

Strengths:
1. This paper is very interesting and well written.
Weaknesses:
1. From the Figure 2, the image results is much better than the text results. Is it necessary to conduct more experiment on more text datasets?

---

> ### Author Response · Authors · 2022-08-02
> **Response**
>
> We thank the reviewer very much for their review! We are glad they found our work very interesting and well written.
>
> While the gains for our Amazon (text) dataset are not as dramatic as some of the image results, we should note that the improvements are still significant in their own right (+2.5% worst-group accuracy, 30% fewer labels). We believe these observations, in tandem with the other extensive experiments in the paper, well support the main takeaway from the paper: AL helps pretrained models learn the intended task. However, we do think a deeper dive into AL for task ambiguity for large language models (or robotics models, speech models, etc) could be a very promising direction for followup work. Another important factor may be that the domain shifts in the image domains are larger or more explicit, whereas the Amazon dataset considers the worst 10% of in-domain reviewers—it is perhaps noteworthy that AL still succeeds here despite this more implicit distribution shift. We'll include this discussion in our next revision.

---

### Meta-Review · Area_Chair_VjL3 · 2022-08-29

**Recommendation:** Accept
**Confidence:** Less certain

**Metareview:**

The paper makes a simple but compelling claim, that using large pretrained models within an active learning loop improves the ability to collect labeled data efficiently for improved model training with limited labeling resources.  All revieweres agreed that the work is technically correct, that the empirical verification was thoroughly done.

The main point of contention among reviewers was whether this idea is novel or surprising enough to warrant publication at NeurIPS.  This is a tricky issue, and I spent considerable time digging into the paper itself after reading through all reviews and author responses.  In the end, I believe that it is important for the field to avoid creating a world in which only sensational-sounding ideas are published.  Yes, it is relatively intuitive that using a learned representation such as a large pretrained model will help put active learning in a lower dimensional space that will be easier to search through, making AL more efficient.  But the field on ML is filled with ideas that sound good in the abstract but fail in practice.  I believe that there is clear value to the field to having rigorous evaluation of obvious good ideas, both as a reality check and also to provide a clear documentation of current state of the art so that the field can build from there.


**Award:**

No

---

### Decision · Program_Chairs · 2022-09-14

Accept